# Protein S is Protective in Acute Lung Injury by Inhibiting Cell Apoptosis

**DOI:** 10.3390/ijms20051082

**Published:** 2019-03-02

**Authors:** Prince Baffour Tonto, Taro Yasuma, Tetsu Kobayashi, Corina N. D’Alessandro-Gabazza, Masaaki Toda, Haruko Saiki, Hajime Fujimoto, Kentaro Asayama, Kentaro Fujiwara, Kota Nishihama, Tomohito Okano, Atsuro Takeshita, Esteban C. Gabazza

**Affiliations:** 1Department of Immunology, Mie University, Graduate School of Medicine Mie, Edobashi 2-174, Tsu, Mie 514-8507, Japan; 316MS02@m.mie-u.ac.jp (P.B.T.); t-yasuma0630@clin.medic.mie-u.ac.jp (T.Y.); t-masa@doc.medic.mie-u.ac.jp (M.T.); johnpaul0114@yahoo.co.jp (A.T.); gabazza@doc.medic.mie-u.ac.jp (E.C.G.); 2Department of Diabetes and Endocrinology, Mie University, Graduate School of Medicine Mie, Edobashi 2-174, Tsu, Mie 514-8507, Japan; kn2480@gmail.com; 3Department of Pulmonary and Critical Care Medicine, Mie University, Graduate School of Medicine Mie, Edobashi 2-174, Tsu, Mie 514-8507, Japan; ktetsu@clin.medic.mie-u.ac.jp (T.K.); mchharuko@city-hosp.matsusaka.mie.jp (H.S.); genfujimoto1974@yahoo.co.jp (H.F.); longwan@yahoo.co.jp (K.A.); kentaro-fu@clin.medic.mie-u.ac.jp (K.F.); okat-omo-525@live.jp (T.O.)

**Keywords:** acute lung injury, protein S, apoptosis, signal pathway, Erk1/2, lipopolysaccharide

## Abstract

Acute lung injury is a fatal disease characterized by inflammatory cell infiltration, alveolar-capillary barrier disruption, protein-rich edema, and impairment of gas exchange. Protein S is a vitamin K-dependent glycoprotein that exerts anticoagulant, immunomodulatory, anti-inflammatory, anti-apoptotic, and neuroprotective effects. The aim of this study was to evaluate whether human protein S inhibits cell apoptosis in acute lung injury. Acute lung injury in human protein S transgenic and wild-type mice was induced by intratracheal instillation of lipopolysaccharide. The effect of human protein S on apoptosis of lung tissue cells was evaluated by Western blotting. Inflammatory cell infiltration, alveolar wall thickening, myeloperoxidase activity, and the expression of inflammatory cytokines were reduced in human protein S transgenic mice compared to the wild-type mice after lipopolysaccharide instillation. Apoptotic cells and caspase-3 activity were reduced while phosphorylation of extracellular signal-regulated kinase was enhanced in the lung tissue from human protein S transgenic mice compared to wild-type mice after lipopolysaccharide instillation. The results of this study suggest that human protein S is protective in lipopolysaccharide-induced acute lung injury by inhibiting apoptosis of lung cells.

## 1. Introduction

Acute lung injury (ALI) and its severe form, acute respiratory distress syndrome (ARDS), are non-cardiogenic pulmonary edema, clinically defined as a severe dysfunction of gas exchange and chest radiographic abnormalities in the absence of heart failure [1,2]. Several conditions including severe sepsis, trauma, and ischemia/reperfusion injury may cause ALI/ARDS [1]. Multiple therapeutic modalities are being used including lung-protective ventilation strategies, prone position and fluid-conservative therapy, but most of them are ineffective in controlling the disease [1,3,4]. The mortality rate of patients with ARDS remains very high ranging between 35% and 46% [3,4,5]. Therefore, development of new therapeutic strategies is urgently required. The pathogenesis of the disease is not completely clear but apoptosis of alveolar epithelial cells is considered to play a critical role [6,7]. Enhanced apoptosis of lung epithelial cells leads to damage and rupture of the pulmonary epithelial/endothelial barrier [8]. Lipopolysaccharide (LPS) that accumulates during sepsis by Gram-negative bacteria is an important cause of lung epithelial cell apoptosis [9,10]. The relevance of cell apoptosis in the mechanism of ALI has been demonstrated by studies showing upregulation of the Fas/FasL system with activation of pro-apoptotic signal pathways in the lung from patients with ALI [7,11]. Inhibition of caspases, which are essential enzymes for the process of cell apoptosis, has been reported to prolong the survival rate in experimental LPS-induced acute lung injury, and inhibition of pro-apoptotic signaling pathways attenuates LPS-induced apoptosis and release of inflammatory cytokines in rat type II alveolar epithelial cells [12,13]. Apart from supporting the role of apoptosis in ALI, these studies point to apoptosis of lung cells as a potential target for therapeutic intervention in ALI/ARDS.

Protein S (PS) is a vitamin K-dependent glycoprotein that has anticoagulant action by enhancing several-fold the inhibitory activity of APC on blood coagulation, by directly blocking the activity of prothrombinase complex, tenase, or by stimulating the inhibition of the tissue factor inhibitory pathway [14,15]. Apart from its anticoagulant activity, PS can also independently exert anti-inflammatory effect [16]. Previously we demonstrated that administration of PS in mice can attenuate cell infiltration and the expression of inflammatory cytokines in LPS-induced ALI [16]. PS also has anti-apoptotic activity [17,18]. PS reduces bleomycin-induced pulmonary fibrosis by inhibiting apoptosis of lung epithelial cells and ameliorates streptozotocin-induced diabetes by suppressing apoptosis of pancreatic β-cells [17,18]. However, the strong anti-apoptotic activity of PS can be also detrimental; for example, previous studies have shown that administration of PS worsens acute liver injury induced by alcohol and chronic liver injury and liver fibrosis caused by carbon tetrachloride [19,20].

In the present study, we hypothesized that overexpression of PS will attenuate ALI not only by its anti-inflammatory activity but also by its inhibitory activity on cell apoptosis. To demonstrate this hypothesis, in this study we evaluated and compared the inflammatory and apoptosis-related markers between wild-type mice and mice overexpressing human PS in the lungs.

## 2. Results

### 2.1. Systemic and Lung Upregulation of Protein S (PS) During Lipopolysaccharide (LPS)-Induced ALI

As expected, the antigen concentration of PS was significantly increased in plasma and lung tissue samples from both hPS-TG/SAL and hPS-TG/LPS mice compared to samples from their WT counterparts. The plasma concentration of PS was significantly higher in the hPS-TG/LPS group than in the hPS-TG/SAL group. The mRNA expression of hPS was positive in both hPS-TG groups but it was not detected in both WT groups (Figure 1). No difference in the mRNA expression of mouse PS was observed between groups.

### 2.2. Less Lung Cell Infiltration in Mice Overexpressing Human Protein S (hPS)

The total number of infiltrating cells and the total number of neutrophils in BALF were significantly increased in WT/LPS and hPS-TG/LPS group compared to WT/SAL and hPS-TG/SAL groups, respectively. The total number of cells and the total number of neutrophils were decreased in the hPS-TG/LPS group compared to the WT/LPS group but the differences were not statistically significant (Figure 2A). There were not significant differences in the count of macrophages and lymphocytes among groups (Figure 2A).

The number of infiltrating inflammatory cells in lung tissue was significantly increased in lung tissue from hPS-TG/LPS and WT/LPS mice compared to mice receiving intratracheal saline, but it was significantly decreased in hPS-TG/LPS mice compared to WT/LPS mice (Figure 2B,C).

### 2.3. The Coagulation System Was Not Affected by hPS Overexpression

No significant difference in the plasma concentration of TAT was found between WT/LPS and hPS-TG/LPS groups or between hPS-TG/SAL and hPS-TG/LPS groups (Figure 3). The BALF concentration of TAT was significantly higher in hPS-TG/LPS group compared to hPS-TG/SAL group but not between WT/SAL and WT/LPS groups. The lung tissue concentration of TAT was significantly increased in WT/LPS and hPS-TG/LPS groups compared to WT/SAL and hPS-TG/SAL groups, respectively (Figure 3). No significant difference in the plasma, BALF and lung tissue concentrations of TAT was observed between WT/LPS and hPS-TG/LPS groups (Figure 3).

### 2.4. Suppression of Pro-Inflammatory Markers by hPS Overexpression

The plasma levels of MCP-1, MPO, TNF-α and IL-6 were significantly higher in WT/LPS mice than in WT/SAL mice but there was no difference between hPS-TG/LPS and hPS-TG/SAL groups. The plasma levels of MCP-1 and MPO were significantly decreased in hPS-TG/LPS group compared to their WT counterpart (Figure 4A).

The BALF levels of MPO, TNF-α and IL-6 were significantly higher in the WT/LPS group than in the WT/SAL group but no difference was observed between hPS-TG/LPS and hPS-TG/SAL groups. The BALF levels of MPO, TNF-α and IL-6 were significantly reduced in the hPS-TG/LPS group compared to its WT counterpart (Figure 4B). The BALF level of MCP-1 was not significantly different between groups.

The lung tissue levels of MCP-1, MPO, TNF-α and IL-6 were significantly increased in the WT/LPS group compared to the WT/SAL group but they were significantly decreased in the hPS-TG/LPS group compared to the WT/LPS group (Figure 4C). No difference was found between hPS-TG/LPS and hPS-TG//SAL groups.

The protein concentration of total protein in BALF was significantly (*p* < 0.05) enhanced in both WT/LPS (339.1 ± 52.0 µg/mL), and hPS-TG/LPS (321.1 ± 84.3 µg/mL) groups compared to the WT/SAL (193.1 ± 37.2 µg/mL) and hPS-TG/SAL (129.0 ± 27.2 µg/mL) groups but there was no significant difference between WT/LPS and hPS-TG/LPS groups.

### 2.5. Decreased Lung Apoptotic Cells in Mice Overexpressing hPS

The number of TUNEL (+) cells in lung tissue was significantly increased in the WT/LPS group compared to the WT/SAL group. There was not significant statistical difference between hPS-TG/LPS and hPS-TG/SAL groups in the number of TUNEL (+) cells. The number of TUNEL (+) cells in lung tissue was significantly increased in WT/LPS group compared to hPS-TG/LPS group (Figure 5A,B).

### 2.6. The Expression of Apoptotic Factors Is Regulated by hPS

The mRNA expression of the markers of apoptosis inhibition Bcl2 and Bcl-xl and both the Bcl2/Bax and Bcl-xl/Bax ratios significantly increased in the hPS-TG/LPS group compared to the WT/LPS group (Figure 6). The mRNA expression of Bcl-xl and both the Bcl2/Bax and Bcl-xl/Bax ratios significantly decreased in the WT/LPS group compared to the WT/SAL group (Figure 6). The mRNA expression of Bcl2 decreased, but not at a statistically significant level, in the WT/LPS group compared to the WT/SAL group (Figure 6).

### 2.7. Decreased Activation of Caspase-3 in Mice Overexpressing hPS

Western blotting of lung tissue showed that the cleaved form of caspase-3, a marker of cell apoptosis, is increased in the WT/LPS group compared to the WT/SAL and hPS-TG/LPS groups. There was no difference between hPS-TG/LPS and hPS-TG/SAL groups (Figure 7A).

The activation of Erk1/2 was significantly increased in the hPS-TG/LPS group compared to the WT/LPS group but no statistical difference was found between other groups (Figure 7B).

## 3. Discussion

The results of the present study show that overexpression of PS in the lungs attenuates ALI by inhibiting apoptosis of lung cells. PS is a 69 KDa glycoprotein expressed by multiple cells including hepatocytes, vascular endothelial cells, lymphocytes and lung cells [14]. Besides its anticoagulant effect, PS can also inhibit the inflammatory response, the complement system, and prolong cell survival. A previous study has shown that administration of hPS ameliorates LPS-induced ALI in mice by suppressing the mRNA and protein expressions of pro-inflammatory cytokines and chemokines from alveolar epithelial cells and macrophages, suggesting the therapeutic benefits of the anti-inflammatory activity of hPS in ALI [16]. Consistent with these findings, here we found significantly decreased inflammatory cells in lung tissue and significantly decreased lung concentrations of MCP-1, TNF-α and IL-6 with reduced release of MPO in mice overexpressing hPS compared to their WT counterparts after inhalation of LPS, further supporting the favorable effect of hPS-mediated inhibition of inflammation. It is worth noting that there was no significant difference in the BALF concentration of protein between WT and hPS-TG treated with LPS, suggesting that hPS overexpression exerts no effect on the epithelial/endothelial barrier. In addition, the beneficial effect of hPS was not due to its anticoagulant effect because the level of TAT, a marker of coagulation activation, was not decreased in the hPS-TG/LPS group compared to the WT/LPS group.

Previous studies have shown that apoptosis of endothelial and alveolar epithelial cells plays a critical role in the pathogenesis of ALI [21,22,23]. Enhanced cell apoptosis may be detrimental because abnormal accumulation of apoptotic cells can perpetuate the inflammatory response by the increased release of inflammatory mediators, proteases, reactive oxygen species or lysozymes [24]. Therefore, inhibition of cell apoptosis may prevent tissue damage, dampen the immune response and accelerate tissue resolution and healing during acute inflammatory conditions including ALI [24]. Previous studies have shown that PS may directly inhibit the cell process of apoptosis by binding to the TAM (Axl, Mer, Tyro3) family of receptor tyrosine kinases. Interaction of PS with TAM receptors activates the signaling pathways of phosphoinosotide 3-kinase(PI3k)/Akt and the mitogen-activated protein kinase/extracellular signal-regulated kinase(MAPK/Erk) that promote cell growth and inhibit cell apoptosis [25,26,27,28]. In addition, we previously demonstrated that hPS inhibits apoptosis by activating the PI3K/Akt pathway in a variety of cells including lung epithelial cells [17,18,19,20]. However, to date whether hPS also suppresses apoptosis of lung cells in ALI remains unknown. In the present study, we demonstrated that the number of lung apoptotic cells and the cleavage of caspase-3 are decreased and the activation of Erk1/2 is increased in mice overexpressing hPS compared to WT after induction of ALI by intratracheal instillation of LPS. In addition, the expression of the anti-apoptotic factors Bcl2, Bcl-xl and the Bcl2/Bax and Bcl2-xl/Bax ratios were significantly enhanced in mice overexpressing hPS compared to their WT counterpart after treatment with LPS. These observations suggest that hPS inhibits apoptosis of lung cells in ALI. It is worth noting here that inhibition of apoptosis by hPS may be detrimental in some acute or chronic diseases in which its target cells play a pathological role. For example, there is evidence showing that hPS may worsen acute alcoholic liver injury by inhibiting apoptosis of activated natural killer T cells, enhance liver fibrosis by prolonging the survival of fibrogenic stellate cells or promote cancer progression and metastasis by providing a survival advantage to cancer cells [19,20,29]. However, beneficial effects of hPS have been reported in animal models of acute and chronic disorders of the lungs including ALI and pulmonary fibrosis [16,17]. The differential expression level of hPS receptors on the cell surface may explain these organ-dependent differential effects of hPS [30].

The low power of the experiment due to the small number of animals allocated in each experimental group is a limitation of the present study and, therefore, the reported statistical differences and related clinical implications should be interpreted with caution. However, this is the first report showing that hPS inhibits apoptosis of lung cells in ALI, thereby suggesting the potential beneficial effect of hPS in this devastating disease. Further studies should be undertaken to corroborate these findings.

## 4. Materials and Methods

### 4.1. Experimental Animals

Characterization of the homozygous human (h) PS transgenic (TG) mouse on a C57BL/6 background has been previously reported [19]. Wild-type (WT) C57BL/6 mice were used as controls. Male mice (8- to 10-week old) were used in the experiments. All animals were maintained in a specific pathogen-free environment and subjected to a 12 h light:dark cycle in the animal house of Mie University. Genotyping of hPS mice was performed by polymerase chain reaction (PCR) using DNA extracted from the tails and by measuring the plasma concentration of PS antigen as previously described [19]. The Committee for Animal Investigation of Mie University approved the protocols(Approval No: 24-50, Date: 2016/02/01) of the study and all animal procedures were performed in accordance with the institutional guidelines of Mie University and following the internationally approved principles of laboratory animal care published by the National Institute of Health (https://olaw.nih.gov/).

### 4.2. Acute Lung Injury (ALI) Induction

Lung damage was induced by intratracheal instillation of LPS (5 mg/kg; Sigma-Aldrich, St. Louis, MO, USA). Mice were categorized into the following groups: WT/LPS (*n* = 7) and hPS-TG/LPS (*n* = 7) groups that received intratracheal instillation of LPS dissolved in physiological saline (SAL) (75 µL), and WT/SAL (*n* = 3) and hPS-TG/SAL (*n* = 3) groups that received intratracheal instillation of saline (75 µL). The instillation of LPS and saline was performed under anesthesia by intraperitoneal injection of 62.5 mg/kg sodium pentobarbital. Mice were sacrificed 24h after LPS instillation by an overdose of intraperitoneal pentobarbital.

### 4.3. Collection of Samples

After mouse euthanasia, bronchoalveolar lavage fluid (BALF) was sampled as previously described and blood samples were collected by heart puncture and placed in tubes containing 10 U/mL heparin [17]. The total number of cells in BALF was measured using a nucleocounter from ChemoMetec (Allerød, Denmark). The BALF supernatant was separated by centrifugation and stored at −80 °C until use for biochemical analysis. For differential cell counting BALF was centrifuged using a cytospin and the cells were stained with May–Grunwald–Giemsa (Merck, Darmstadt, Germany).

### 4.4. Histological Examination

Mice were thoracotomized under profound anesthesia, the pulmonary circulation flushed with saline and then the lungs were excised. One of the lungs was perfused with 10% neutral buffered formalin, fixed in formalin for 24 h and then embedded in paraffin. Tissue specimens of 5 µm were prepared, stained with hematoxylin-eosin and then examined under a light microscopy (Olympus BX53 microscope; Tokyo, Japan). An investigator blinded to the treatment group counted the number of inflammatory cells in a blinded fashion; at least five microscopic fields were counted per mouse. Cell counting in the histological sections was performed using the Olympus BX53 microscope with a plan objective, combined with an Olympus DP70 digital camera (Tokyo, Japan) and the WinROOF image processing software (Mitani Corp., Fukui, Japan) for Windows.

### 4.5. Western Blotting

Activation of ERK in lung tissue was evaluated by Western blotting following standard methods using specific antibodies purchased from Cell Signaling (Danvers, MA, USA). Apoptosis was evaluated by measuring the cleaved form of caspase-3 (Cell Signaling) by Western blotting using specific antibodies.

### 4.6. Immunohistochemistry

DNA fragmentation in lung tissue was evaluated by the terminal deoxynucleotidyl transferase-mediated dUTP nick end labeling (TUNEL) method. TUNEL staining was performed at the Biopathology institute Corporation using terminal deoxynucleotidyl transferase enzyme (Millipore Sigma, St. Louis, MO, USA), anti-digoxigenin peroxidase (Millipore Sigma, St. Louis, MO, USA) and 3,3′-diaminobenzidine. The total number of TUNEL (+) cells was counted using the WinROOF image processing software (Mitani Corporation, Osaka, Japan).

### 4.7. Biochemical Analysis

Lung tissue was homogenized in the presence of a protease inhibitor cocktail (Nacalai Tesque Inc.; Kyoto, Japan) using Tomy Micro Smash™ MS-100R (Tomy Digital Biology Co., Tokyo, Japan), the preparation was then centrifuged at 15,000 rpm for 10 min and supernatants collected and stored at −80 °C until analysis. The concentration of PS was measured using polyclonal antihuman PS antibody (Dako, Santa Clara, CA, USA) as capture antibody and biotin-labeled monoclonal antihuman PS antibody (Haematologic Technologies Inc., Esset, VT) as second antibody; absolute values were extrapolated from a standard curve drawn using PS antigen (Enzyme Research Laboratories, South Bend, IN, USA). The detection limit of this PS immunoassay is 0.8 ng/mL. The polyclonal anti-hPS antibody used for this immunoassay cross-reacts with mouse PS [19]. The concentration of total protein was measured using a commercial kit (BCATM protein assay kit; Pierce, Rockford, IL, USA) following the manufacturer’s instructions. The concentrations of TNF-α, IL-6 and MCP-1 were measured using commercial enzyme immunoassay (EIA) kits from BD Biosciences Pharmingen (San Diego, CA, USA) following the manufacturer’s instructions. Thrombin–antithrombin complex (TAT), a marker of coagulation activation, was measured using enzyme immunoassay kits from Cedarlane Laboratories (Hornby, ON, Canada) following the manufacturer’s instructions. The concentration of myeloperoxidase (MPO) was measured by a colorimetric assay using the synthetic substrate 2,2′-azinobis (ethylbenzyl-thiazoline-6-sulfonic acid) diammonium salt.

### 4.8. Gene Expression Analysis

Total RNA was harvested from the whole lung tissue with TRIzol reagent (Invitrogen, Carlsbad, CA), according to the manufacturer’s instructions. The first-strand cDNA was synthesized from 2 µg of total RNA with oligo-dT primer and SuperScript II RNase H Reverse Transcriptase (Invitrogen), and PCR was performed using gene-specific primers. The RNA concentration was measured by ultraviolet (UV) absorption at 260:280 nm using an Ultrospec 1100 pro ultraviolet/visible (UV/Vis) spectrophotometer (Amersham Biosciences, NJ, USA). Reverse transcription of RNA into cDNA was done using a ReverTra Ace quantitative reverse transcription PCR (qPCR RT) kit (TOYOBO, Osaka, Japan) and then the DNA was amplified by PCR using Quick Taq HS DyeMix (TOYOBO). The primers used in the experiments were as follows: mouse glyceraldehyde 3-phosphate dehydrogenase (GAPDH), forward: 5′-CCCTTATTGACCTCAACTACATGGT-3′, reverse: 5′-GAGGGGCCATCCACAGTCTTCTG-3′; Mouse PS, forward: 5′-TGCTCAGTTCAGCATAGCTACA-3′, reverse: 5′-CTGATCCGAGCACAGAGATACC-3′; human PS, forward: 5′-AGGGCTCCTACTATCCTGGTTCTG-3′, reverse: 5′-GCCATTATAAAAGGCATTCACTGG-3′; B-cell lymphoma2 (Bcl2), forward: 5′-AGCTGCACCTGACGCCCTT-3′, reverse: 5′-GTTCAGGTACTCAGTCATCCAC-3′; Bcl-extra large (Bcl-xl), forward: 5′-AGGTTCCTAAGCTTCGCAATTC-3′, reverse: 5′-TGTTTAGCGATTCTCTTCCAGG-3′; Bcl2-associated x (Bax), forward: 5′-CGGCGAATTGGAGATGAACTG-3′, reverse: 5′-GCAAAGTAGAAGAGGGCAACC-3′; as described [19]. The gene expression was normalized by the transcription level of glyceraldehyde-3-phosphate dehydrogenase.

### 4.9. Statistical Analysis

Data are expressed as the mean ± standard deviation of the means (S.D.). The statistical difference between variables was calculated by analysis of variance with post hoc analysis using the Fisher’s least significant difference test. Normality of the data was calculated using the W/S test [30]. Statistical analyses were performed using the StatView 4.1 package software for the Macintosh (Abacus Concepts, Berkeley, CA, USA). Statistical significance was considered as *p* < 0.05.

## Figures and Tables

**Figure 1 ijms-20-01082-f001:**
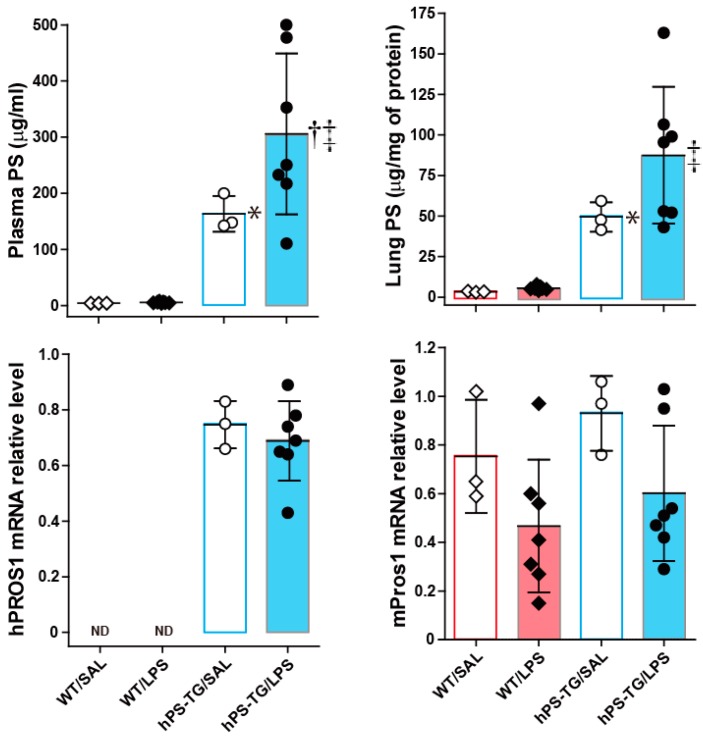
Systemic and local upregulation of Protein S (PS) during lipopolysaccharide (LPS)-induced acute lung injury (ALI). Mice were allocated in four groups including wild type mice treated with saline (WT/SAL, *n* = 3) or LPS (WT/LPS, *n* = 7), and human PS transgenic (TG) mice treated with saline (hPS TG/SAL, *n* = 3) or LPS (hPS TG/LPS, *n* = 7). The level of PS was measured by enzyme immunoassay and the gene expression by reverse transcription polymerase chain reaction (RT-PCR). Data are expressed as the mean ± SD. * *p* < 0.05: hPS-TG/SAL vs WT/SAL; † *p* < 0.05: hPS-TG/LPS vs hPS-TG/SAL; ‡ *p* < 0.05: hPS-TG/LPS vs WT/LPS. ND, not detected.

**Figure 2 ijms-20-01082-f002:**
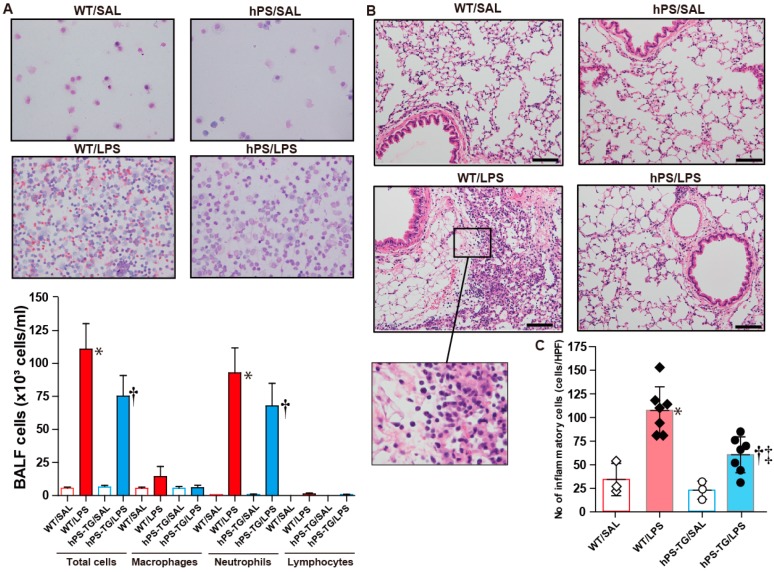
Less lung cell infiltration in mice overexpressing hPS. Mice were allocated in four groups including wild type mice treated with saline (WT/SAL, *n* = 3) or LPS (WT/LPS, *n* = 7), and human PS transgenic (TG) mice treated with saline (hPS TG/SAL, *n* = 3) or LPS (hPS TG/LPS, *n* = 7). Cells in bronchoalveolar lavage fluid were counted using automatic cell counter and stained for differential counting (**A**). Lung tissue samples were stained with hematoxylin & eosin (**B**). The number of cells was counted using the WindROOF image processing software (**C**). Scale bars indicate 100 µm. Data are expressed as the mean ± SD. * *p* < 0.05: WT/LPS vs WT/SAL; † *p* < 0.05: hPS-TG/LPS vs hPS-TG/SAL; ‡ *p* < 0.05: hPS-TG/LPS vs WT/LPS.

**Figure 3 ijms-20-01082-f003:**
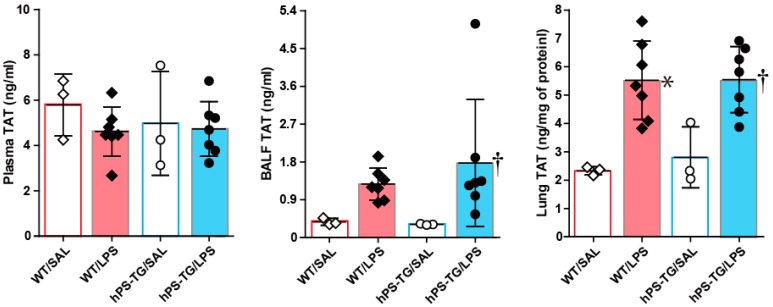
Activation of the coagulation system was inhibited by hPS overexpression. Mice were allocated in four groups including wild type mice treated with saline (WT/SAL, *n* = 3) or LPS (WT/LPS, *n* = 7), and human PS transgenic (TG) mice treated with saline (hPS TG/SAL, *n* = 3) or LPS (hPS TG/LPS, *n* = 7). The concentration of thrombin–antithrombin (TAT) was measured using an enzyme immunoassay. Data are expressed as the mean ± SD. * *p* < 0.05: WT/LPS vs WT/SAL; † *p* < 0.05: hPS-TG/LPS vs hPS-TG/SAL.

**Figure 4 ijms-20-01082-f004:**
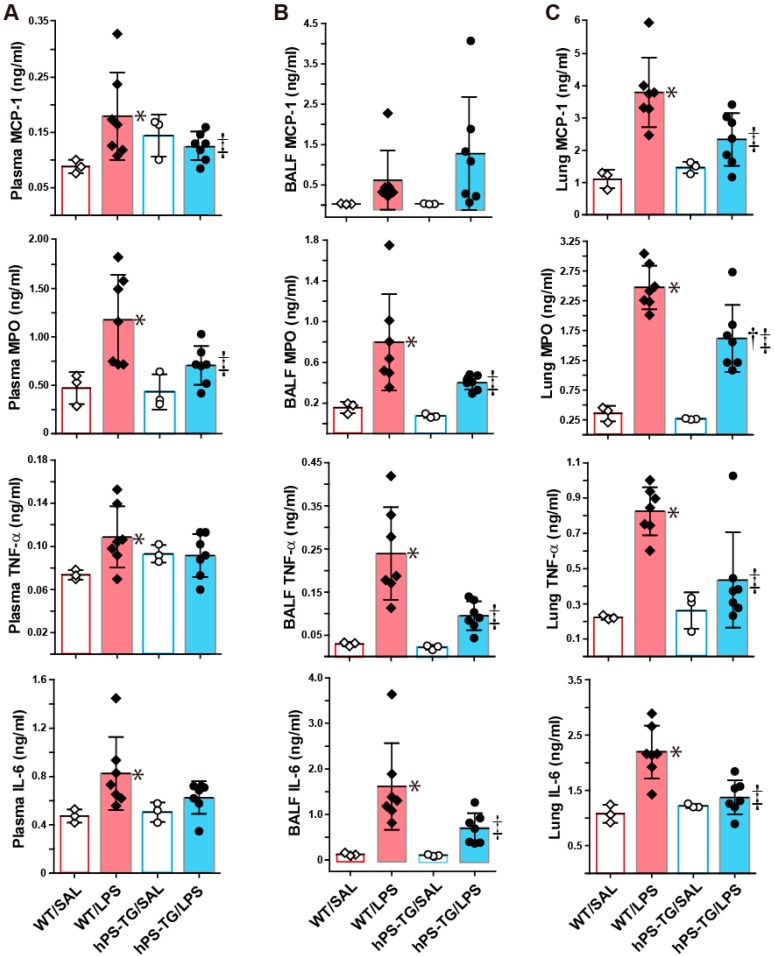
Suppression of pro-inflammatory markers by hPS overexpression. Mice were allocated in four groups including wild type mice treated with saline (WT/SAL, *n* = 3) or LPS (WT/LPS, *n* = 7), and human PS transgenic (TG) mice treated with saline (hPS TG/SAL, *n* = 3) or LPS (hPS TG/LPS, *n* = 7). The plasma (**A**), bronchoalveolar lavage fluid (**B**) and lung tissue (**C**) concentrations of cytokines and chemokines were measured using enzyme immunoassays and myeloperoxidase was measured using a colorimetric assay as described under materials and methods. Data are expressed as the mean ± SD. * *p* < 0.05: WT/LPS vs WT/SAL; † *p* < 0.05: hPS-TG/LPS vs hPS-TG/SAL; ‡ *p* < 0.05: hPS-TG/LPS vs WT/LPS.

**Figure 5 ijms-20-01082-f005:**
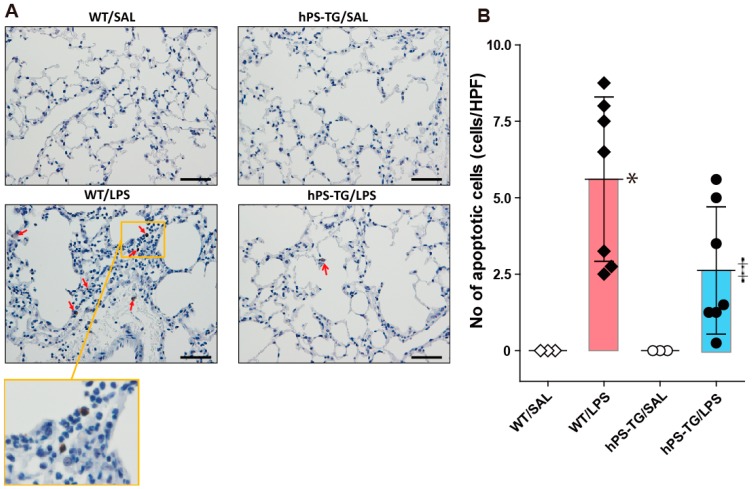
Decreased lung apoptotic cells in mice overexpressing hPS. Mice were allocated in four groups including wild type mice treated with saline (WT/SAL, *n* = 3) or LPS (WT/LPS, *n* = 7), and human PS transgenic (TG) mice treated with saline (hPS TG/SAL, *n* = 3) or LPS (hPS TG/LPS, *n* = 7). Staining of terminal deoxynucleotidyl transferase dUTP nick-end labeling (TUNEL) was performed as described under materials and methods (**A**) and the number of apoptotic cells was counted using WindROOF image processing software (**B**). Arrows indicate apoptotic cells. Scale bars indicate 100 µm. Data are expressed as the mean ± SD. **p* < 0.05: WT/LPS vs WT/SAL; ‡ *p* < 0.05: hPS-TG/LPS vs WT/LPS.

**Figure 6 ijms-20-01082-f006:**
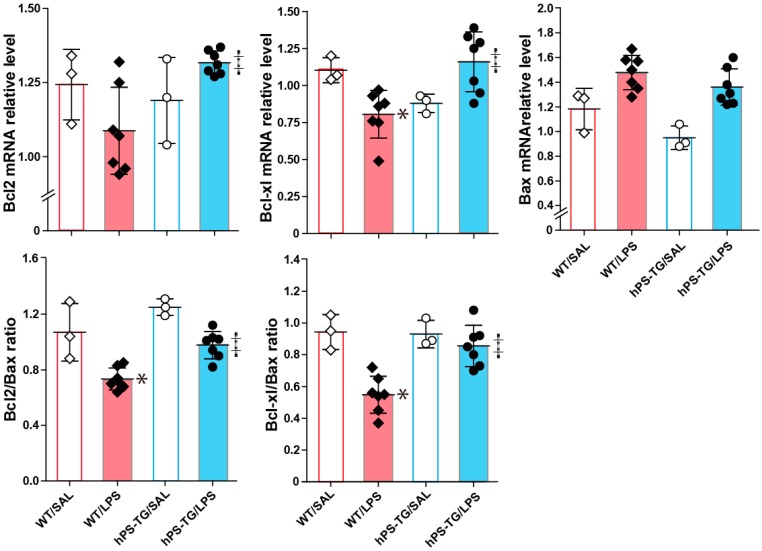
Regulation of the expression of apoptotic factors by hPS. Mice were allocated in four groups including wild-type mice treated with saline (WT/SAL, *n* = 3) or LPS (WT/LPS, *n* = 7), and human PS transgenic (TG) mice treated with saline (hPS TG/SAL, *n* = 3) or LPS (hPS TG/LPS, *n* = 7). Total RNA was extracted from the whole lung tissue, reverse-transcribed and then PCR was performed using gene-specific primers. The mRNA expression of the anti-apoptotic factors Bcl2 and Bcl-xl and of the apro-apoptotic factor was evaluated by PCR. Data are expressed as the mean ± SD. **p* < 0.05: WT/LPS vs WT/SAL; ‡ *p* < 0.05: hPS-TG/LPS vs WT/LPS.

**Figure 7 ijms-20-01082-f007:**
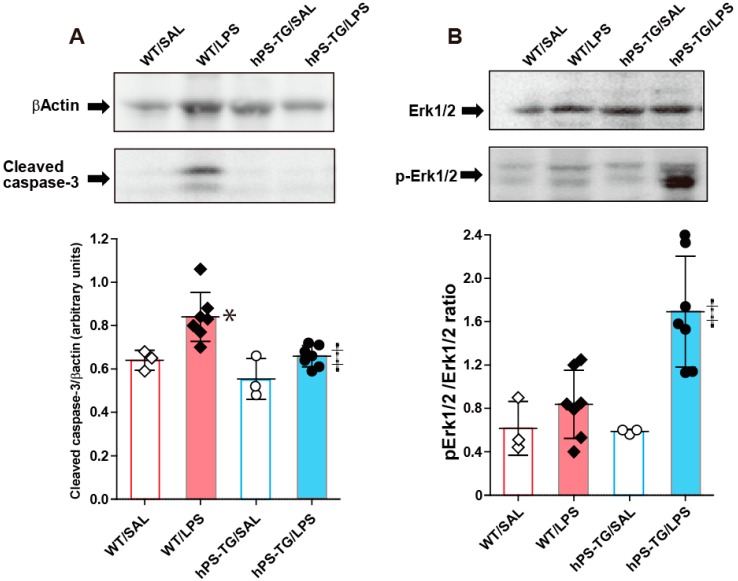
Decreased cleavage of caspase-3 and increased activation of Erk1/2 in mice overexpressing hPS. Mice were allocated in four groups including wild-type mice treated with saline (WT/SAL, *n* = 3) or LPS (WT/LPS, *n* = 7), and human PS transgenic (TG) mice treated with saline (hPS TG/SAL, *n* = 3) or LPS (hPS TG/LPS, *n* = 7). The presence of cleaved caspase-3 (**A**) and p-Erk1/2 (**B**) in lung tissue was evaluated by Western blotting. Quantification was undertaken using ImageJ (https://imagej.nih.gov/ij/index.html). Data are expressed as the mean ± SD. * *p* < 0.05: WT/LPS vs WT/SAL; ‡ *p* < 0.05: hPS-TG/LPS vs WT/LPS. p-Erk1/2, phosphorylated-Erk1/2.

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
