# Peer review of "Protein S is Protective in Acute Lung Injury by Inhibiting Cell Apoptosis"

_ijms, 2019, doi:10.3390/ijms20051082_

Round 1

Reviewer 1 Report

There is some improvement in this iteration of the manuscript in that some numbers of rodents has been increased in LPS treated groups. In addition, the group has provided additional experiments to support the role of diminished apoptosis by TG rodents over-expressing PS. From this standpoint, the work is better.  

Typographical errors have not all been corrected.  For example in figure 2, scale bar is misspelled.

There are still statistical questions regarding the data.  In figure two, the legend indicates that †p<0.05 vs hPSTG/SAL; ‡p<0.05 vs WT/LPS.   However, the single symbol in the total number of lavage cells graph sits above the bar for hPSTG LPS, so it is not possible to tell what the comparison is.  The real question is are the hPS TG groups treated with LPS different than WT treated with the same. The text says “The total number of cells and the total number of neutrophils tended to be lower in the hPSTG/LPS group than in the WT/LPS group”, but what does “tended” mean?  Is it significant or not?

The question about normality of data distribution does not appear to have ben answered. Thus all data are presented as means and SDs.

Both pro- (BCl2 and BClx) and anti- (BAX) apoptotic RNA in whole lung are elevated in this model, but the results section (or figure legends) do not state that RNA were extracted from whole lung tissue. This omission should be corrected.

Author Response

RESPONSE TO COMMENTS OF REVIEWER #1

COMMENT 1

There is some improvement in this iteration of the manuscript in that some numbers of rodents has been increased in LPS treated groups. In addition, the group has provided additional experiments to support the role of diminished apoptosis by TG rodents over-expressing PS. From this standpoint, the work is better.

RESPONSE

We thank very much the favorable comments of the Reviewer on the revised version of the manuscript.

COMMENT 2

Typographical errors have not all been corrected. For example in figure 2, scale bar is misspelled.

RESPONSE

In addition to errors pointed out by the Reviewer we have corrected other errors found.

Please see lines 27, 83, 84, 116, 216, 317, 318, 380 and 416 in the current revised version of the manuscript.

COMMENT 3

There are still statistical questions regarding the data. In figure two, the legend indicates that †p<0.05 vs hPSTG/SAL; ‡p<0.05 vs WT/LPS. However, the single symbol in the total number of lavage cells graph sits above the bar for hPSTG LPS, so it is not possible to tell what the comparison is.

RESPONSE

We are sorry for the lack of clarity in the meaning of the symbols. We have specified what comparison each symbol is showing. Please see the legend of Figure 1: lines 184, 185, Figure 2: lines 217, 218; Figure 3: lines 242, 243, Figure 4: lines 291, 29w2, Figure 5: line 319, Figure 6: lines 349, 350, and Figure 7: lines 382, 383.

 COMMENT 4

The real question is are the hPS TG groups treated with LPS different than WT treated with the same. The text says “The total number of cells and the total number of neutrophils tended to be lower in the hPSTG/LPS group than in the WT/LPS group”, but what does “tended” mean?  Is it significant or not?

RESPONSE

In the current revised version we clarified whether there was or not significant difference between groups. Please see lines 189 to 191, lines 323 to 326, and lines 392 to 394 in the current revised version of the manuscript.

COMMENT 5

The question about normality of data distribution does not appear to have been answered. Thus all data are presented as means and SDs.

RESPONSE

Because the small number of samples we presented all data as means, SDs and dot plots.

Also, because the small number of sample we used the W/S test to calculate normal distribution and added a new reference (No 21: Kanji, G. K.; SAGE Research Methods Core., 100 Statistical Tests. In 3rd ed.; SAGE Publications, Limited SAGE Publications, Incorporated Distributor: London, Thousand Oaks, 2006; p 256 p).

Please see line 160 and lines 499 and 500.

COMMENT 6

Both pro- (BCl2 and BClx) and anti- (BAX) apoptotic RNA in whole lung are elevated in this model, but the results section (or figure legends) do not state that RNA were extracted from whole lung tissue. This omission should be corrected.

RESPONSE

As suggested by the Reviewer the omitted expression was added in the text. Please see the method section line 138, and Figure 6 legend lines 346 to 348.

Reviewer 2 Report

I am happy with the authors' responses to concerns

Author Response

RESPONSE TO COMMENT OF REVIEWER #2

COMMENT 1

I am happy with authors' responses to concerns

RESPONSE

We thank very much the Reviewer for the favorable consideration on the revised manuscript.

This manuscript is a resubmission of an earlier submission. The following is a list of the peer review reports and author responses from that submission.

Round 1

Reviewer 1 Report

Tonto et al report that over-expression of human protein S (PS) in mice protects against acute lung injury induced by LPS based on diminished inflammatory cell infiltration, alveolar wall thickening, myeloperoxidase and selected inflammatory cytokines.  They also described diminished apoptosis based on cleaved caspase 3 protein density.  On the strength of these data, the authors claim pulmonary over-expression of PS is protective in acute lung injury because of diminished apoptosis and diminished inflammatory responses.

While the role of protein in lung injury may be an interesting question, there are several serious problems with the approach and data in this manuscript which limit potential conclusions.

Some (many?) statistics appear to be been performed with only n of 3 in WT and TG saline groups.  The power of these experiments in not adequate with such numbers.

The “n”s for each experiment should be provided.  They are included only sporadically.

Differences in cleaved caspase 3 protein are claimed between WT-LPS and TG-LPS mice (figure 5).  However, with an “n” of 4 in each group (?) and an apparent difference in the mean of 20-25% and variance in the individual values, the clinical implications and even statistical significance of these values should be questioned.

If apoptosis is an important endpoint for protection by hPS over-expression, at least one or two more markers of apoptosis should be provided. Furthermore, IHC should be performed with lung tissue to identify cell types protected from apoptosis.

The experiment which shows increased phospho-ERK/ERK (figure 5) shows no increase in phospho-ERK, but an increase in ERK.  These data are not discussed at all.  How are they interpreted?

Figure 1 shows no differences in the PMNs or total cell numbers in the lavage between WT-LPS and TG-LPS groups.  These data are not discussed.  How do the authors interpret them?

For many of the cytokine values in figure 4, there is an apparently large overlap in the SD between WT LPS and TG LPS (e.g. lung MPO, plasma MCP-1). And yet the figures suggest statistical significance between these values.  The stats should be reviewed.

The authors attribute protection from LPS-induced lung injury to over-expression of pulmonary hPS. However, it may easily be that over-expression of hPS in neutrophils that modifies the pulmonary LPS response. The experimental design does not permit assessment of the contribution of individual cell types to the response.  

The incremental contribution of the data in this manuscript relative to those already published which show hPS ameliorates LPS induced lung injury in mice with diminished inflammatory cytokines from epithelial cells and macrophages is small.    

Minor concerns:

Figure 1 shows plasma and lung concentrations of PS.  Presumably the lung values represent concentrations per wet lung weight in crude lung homogenate, but there is no description in the methods of how the samples were handled. 

There are several typographical errors throughout the manuscript which should be corrected.

Author Response

RESPONSE TO REVIEWERS’ QUESTIONS

REVIEWER #1

 COMMENT 1

Tonto et al report that over-expression of human protein S (PS) in mice protects against acute lung injury induced by LPS based on diminished inflammatory cell infiltration, alveolar wall thickening, myeloperoxidase and selected inflammatory cytokines. They also described diminished apoptosis based on cleaved caspase 3 protein density. On the strength of these data, the authors claim pulmonary over-expression of PS is protective in acute lung injury because of diminished apoptosis and diminished inflammatory responses. While the role of protein in lung injury may be an interesting question, there are several serious problems with the approach and data in this manuscript which limit potential conclusions.

Response

We thank very the comments of the reviewers that have substantially improved the quality of the manuscript.

COMMENT 2

Some (many?) statistics appear to be been performed with only n of 3 in WT and TG saline groups. The power of these experiments in not adequate with such numbers.

Response

We agree with reviewer regarding the low power of the experiment due to the small nuber of mice. We have added as a limitation of the present study a statement regarding the lower power pointed by the reviewer in the revised version of the manuscript. Please see lines No 372 to 373 in the revised version of the manuscript.

COMMENT 3

The “n”s for each experiment should be provided.  They are included only sporadically.

Response

We have added the number of mice to the legend for each figure. Please see lines 171-172, lines 201-202, 228-230, 274-275, 303-304, and lines 332-335 in the revised version.

COMMENT 4

Differences in cleaved caspase 3 protein are claimed between WT-LPS and TG-LPS mice (figure 5). However, with an “n” of 4 in each group (?) and an apparent difference in the mean of 20-25% and variance in the individual values, the clinical implications and even statistical significance of these values should be questioned.

Response

We understand the low power. For this reason, we added as limitation of the study, a statement as described above. Please see lines 372 to 373.

COMMENT 5

If apoptosis is an important endpoint for protection by hPS over-expression, at least one or two more markers of apoptosis should be provided. Furthermore, IHC should be performed with lung tissue to identify cell types protected from apoptosis. 

Response

We have conducted an immunohistochemistry study by TUNEL method to evaluate DNA fragmentation. Please see the description of the methods in lines 106 to 112, the description of results in lines 281 to 308, and Figure 5.

COMMENT 6

The experiment which shows increased phospho-ERK/ERK (figure 5) shows no increase in phospho-ERK, but an increase in ERK.These data are not discussed at all.  How are they interpreted?

Response

There was typo error in the original manuscript submitted. We have corrected in the revised version. Please see lines 316 to 322 and Figure 6. We apologize for this error.

COMMENT 7

Figure 1 shows no differences in the PMNs or total cell numbers in the lavage between WT-LPS and TG-LPS groups.  These data are not discussed.  How do the authors interpret them?

Response

We evaluated the number of inflammatory cells in lung tissue from each group of mice and found significant difference between WT/LPS and hPS-TG/LPS mice. We have also added the number of macrophages and lymphocytes to show that the main infiltrating cells were neutrophils. Please see lines 178 to 184 and Figure 2.

COMMENT 8

For many of the cytokine values in figure 4, there is an apparently large overlap in the SD between WT LPS and TG LPS (e.g. lung MPO, plasma MCP-1). And yet the figures suggest statistical significance between these values. The stats should be reviewed.

Response

The statistical test was reviewed and found some statistical differences using the Fisher's Least Significant Difference (LSD) test.

COMMENT 9

The authors attribute protection from LPS-induced lung injury to over-expression of pulmonary hPS. However, it may easily be that over-expression of hPS in neutrophils that modifies the pulmonary LPS response. The experimental design does not permit assessment of the contribution of individual cell types to the response.

Response

We completely agree with the Reviewer’s comment on the mechanism of PS protective activity. As pointed out by the Reviewer, there is the possibility of disease-modification by over-expression of hPS in neutrophils, but unfortunately, we have not so far evaluated whether neutrophils express hPS.

COMMENT 10

The incremental contribution of the data in this manuscript relative to those already published which show hPS ameliorates LPS induced lung injury in mice with diminished inflammatory cytokines from epithelial cells and macrophages is small.

Response

We think that clarification of additional mechanisms of the beneficial effect of hPS on ALI could further support its potential therapeutic application in the future.

COMMENT 11

Figure 1 shows plasma and lung concentrations of PS. Presumably the lung values represent concentrations per wet lung weight in crude lung homogenate, but there is no description in the methods of how the samples were handled.  

Response

As suggested the expression of PS in Figure 1 was corrected. Please see Figure 1 from lines 153 to 175.

Reviewer 2 Report

Tonto and colleagues investigated the impact of human protein S, a vitamin K-dependent glycoprotein, expression in mice during exposure to LPS. Authors observed reduced levels of apoptosis, inflammation and ERK phosphorylation. This paper addresses an important topic that requires further exploration and discussion. The majority of the data is well presented in the figures. However, in its present state, the manuscript requires editing, especially with methodology description and data discussion. There are several items that require addressing. These are itemized below, not in order of significance, but in order of presentation.

Major:

1.       Figure 1A-B, the basal levels of PS and hPROS1 are in the low ug/ml range, which is usually the detection limit of ELISA assays. It would be useful to indicate the limit of detection of the assay by a horizontal dotted line on the bar graph. Also, are there detectable levels of human PS or hPROS1 in wild type animals for Figure 1? If not, it would be better to have “not detected” noted here rather than including dots in those groups.

2.       For Figure 2B, the cells accumulating around the airways appear to be mononuclear. Authors have differentiated total cells into neutrophil numbers only. But can other cell types be quantified here? Perhaps an inflammatory score on the tissue would be useful to quantify the inflammation on the H&E staining

3.       Overall, the methodology, results, and discussion text are very limited. Please expand on all methods to give complete details and outline results in greater detail, outlining fold differences and p values. Figure 3, 4 and 5 are not referenced in the text. Authors need to directly reference these figures and give a detailed description of the data in the results section. The discussion needs to be expanded and discussed in greater depth

4.       In figure 5, I think there is a labeling typo for the ERK blots. It appears that they are labeled the wrong way around? Also, ERK1/2 usually has 2 bands but one of the blots has only 1 band. Please replace with another blot to better represent the quantified data. How was data quantified here?

5.       Increased levels of TAT in the BALF suggests that there could be epithelial/endothelial barrier disruption. Can authors tabulate protein levels in the BALF?

Minor:

1.       For Figure 3, I would recommend presenting data as ug/ml and use less y-axis tick intervals to make the graphs less busy. Also, it would be best to divide figure 3 into subsections (e.g. 3A, 3B, 3C, etc) as there are many graphs and would aid the reader when following the text

2.       Please use correct symbols when labeling cytokines, i.e. TNFalpha or TNFα rather than TNFa

3.       Figure 5 has a typo, I presume it should say beta or β Actin on the y-axis. Please also include the y-axis unit label for each graph, i.e. densitometry units, pixel intensity or arbitrary units?

Author Response

RESPONSE TO REVIEWERS’ QUESTIONS

REVIEWER #2

 COMMENT 1

Tonto and colleagues investigated the impact of human protein S, a vitamin K-dependent glycoprotein, expression in mice during exposure to LPS. Authors observed reduced levels of apoptosis, inflammation and ERK phosphorylation. This paper addresses an important topic that requires further exploration and discussion. The majority of the data is well presented in the figures. However, in its present state, the manuscript requires editing, especially with methodology description and data discussion. There are several items that require addressing. These are itemized below, not in order of significance, but in order of presentation.

Response

We thank very the comments of the reviewers that have substantially improved the quality of the manuscript.

COMMENT 2

Figure 1A-B, the basal levels of PS and hPROS1 are in the low ug/ml range, which is usually the detection limit of ELISA assays. It would be useful to indicate the limit of detection of the assay by a horizontal dotted line on the bar graph. Also, are there detectable levels of human PS or hPROS1 in wild type animals for Figure 1? If not, it would be better to have “not detected” noted here rather than including dots in those groups.

Response

The polyclonal antibody we used as primary antibody can recognize both the human and murine PS. Therefore, this EIA can also measure PS in tissues from WT mice. However, because the PS-TG mice have very high levels it appears that the levels in WT mice are almost 0 in Figure 1. But, actually, there are PS levels at nanogram levels. That the primary antibody cross-reacts with human and mouse PS was mentioned in the revised manuscript. Please lines 121 to 122.

Because the detection limit of PS by this EIA is 0.8 ng/ml it would difficult to indicate exactly the detection limit in the graph using dotted lines. Therefore, we preferred to write the detection limit in the method section. Please see line 121 of the revised manuscript.

As pointed out there was no detection of mRNA of hPROS1 in WT mice thus, as suggested, in this case we indicated by ND (not detected) in the figure. Please see Figure 1 from lines 153 to line 175 in the revised manuscript.

COMMENT 3

For Figure 2B, the cells accumulating around the airways appear to be mononuclear. Authors have differentiated total cells into neutrophil numbers only. But can other cell types be quantified here? Perhaps an inflammatory score on the tissue would be useful to quantify the inflammation on the H&E staining

Response

As suggested by the Reviewer, we added the count of macrophages and lymphocytes in the figure. Please see Figure 2A, lines 186 to 206. The number of inflammatory in the lung tissue was also counted as suggested. Please see Figure 2B and C, lines 186 to 199.

COMMENT 4

Overall, the methodology, results, and discussion text are very limited. Please expand on all methods to give complete details and outline results in greater detail, outlining fold differences and p values. Figure 3, 4 and 5 are not referenced in the text. Authors need to directly reference these figures and give a detailed description of the data in the results section. The discussion needs to be expanded and discussed in greater depth.

Response

The methods were expanded as suggested. Please see lines 78 to 85, lines 98 to 99, lines 106 to 112, lines 114 to 122.

The results were also expanded. Please see lines 149 to 150, lines 178 to 184, lines 208 to 214, lines 234 to 315.

The discussion was also expanded. Please see lines 346 to 347, lines 350 to 354, lines 356 to 360, and lines 368 to 375

COMMENT 5

In figure 5, I think there is a labeling typo for the ERK blots. It appears that they are labeled the wrong way around? Also, ERK1/2 usually has 2 bands but one of the blots has only 1 band. Please replace with another blot to better represent the quantified data. How was data quantified here?

Response

We corrected the typo in the Erk blot. Quantification was done using ImageJ as described. Please line 335. Please also see Figure 6 lines 316 to 337.

COMMENT 6

Increased levels of TAT in the BALF suggests that there could be epithelial/endothelial barrier disruption. Can authors tabulate protein levels in the BALF?

Response

The protein concentration was measured in BALF as suggested. Please see lines 247 to 250 in the revised manuscript. There was not significant difference between WT/LPS and hPS-TG/LPS groups.

COMMENT 7

For Figure 3, I would recommend presenting data as ug/ml and use less y-axis tick intervals to make the graphs less busy. Also, it would be best to divide figure 3 into subsections (e.g. 3A, 3B, 3C, etc) as there are many graphs and would aid the reader when following the text

Response

As suggested, we present data as ng/ml instead of ug/ml and used less y-axis tick intervals. We also divided Figure 4 in A, B, C as suggested.

COMMENT 8

Please use correct symbols when labeling cytokines, i.e. TNFalpha or TNFα rather than TNFa

Response

This type was corrected as suggested. Please see lines

COMMENT 9

Figure 5 has a typo, I presume it should say beta or β Actin on the y-axis. Please also include the y-axis unit label for each graph, i.e. densitometry units, pixel intensity or arbitrary units?

Response

The typo was corrected. The unit was included as suggested. Please see Figure 6.

Round 2

Reviewer 1 Report

The revisions in this manuscript are limited in response to major concerns that were raised.

Incrementally new observations are presented. 

Reviewer 2 Report

Authors have address concerns